**COMMUNICATIONS**

# The dynamics of filament assembly define cytoskeletal network morphology

Giulia Foffano[1], Nicolas Levernier[2] & Martin Lenz[1]

The actin cytoskeleton is a key component in the machinery of eukaryotic cells, and it self-assembles out of equilibrium into a wide variety of biologically crucial structures. Although the molecular mechanisms involved are well characterized, the physical principles governing the spatial arrangement of actin filaments are not understood. Here we propose that the dynamics of actin network assembly from growing filaments results from a competition between diffusion, bundling and steric hindrance, and is responsible for the range of observed morphologies. Our model and simulations thus predict an abrupt dynamical transition between homogeneous and strongly bundled networks as a function of the actin polymerization rate. This suggests that cells may effect dramatic changes to their internal architecture through minute modifications of their nonequilibrium dynamics. Our results are consistent with available experimental data.

[1] LPTMS, CNRS, Université Paris-Sud, Université Paris-Saclay, 91405 Orsay, France. [2] Laboratoire de Physique Théorique de la Matière Condensée, UPMC, CNRS UMR 7600, Sorbonne Universités, 4 Place Jussieu, 75252 Paris Cedex 05, France. Correspondence and requests for materials should be addressed to M.L. (email: martin.lenz@u-psud.fr).

The cytoskeleton of living cells is an extremely dynamic system, of which actin is a vital component. Actin monomers are continuously assembled during polymerization; at the same time, actin filaments are bundled together by crosslinkers to form a large variety of structures. Tight bundles thus appear in filopodia, in stress fibres and in the contractile ring involved in cell division, whereas homogeneous networks are found in the cell cortex and in the lamella[1]. Understanding the fundamental mechanisms that determine the formation and the morphology of the actin cytoskeleton is thus necessary to explain how the cell regulates its own shape, internal structure and motility.

A tool of choice to characterize these structures is to isolate a few essential ingredients and study the result of their interactions. Bottom-up experiments thus put one type of crosslinker in solution together with purified actin, resulting in *in vitro* reconstituted actin networks[2,3]. As filaments grow and become crosslinked into bundles, the morphology of the resulting networks strongly depends on their assembly kinetics: different protocols leading to the same filament number and lengths through different kinetic pathways thus result in different structures, demonstrating that the observed phases are out of equilibrium[4–6]. The structure of keratin networks similarly results from the competition between filament elongation and bundle formation[7]. Despite the highly dynamic nature of these experiments, theoretical attempts to describe such networks have largely relied on equilibrium physics[8–10], modelling morphological transitions as the result of the competition between crosslinker binding and thermal fluctuations. In many cases, however, the bundling of two or more filaments by hundreds of crosslinkers can involve energies of the order of thousands of $k_BT$[11,12], indicating that equilibrium thermal fluctuations alone cannot account for the presence of structural disorder in the final networks.

Here we propose a theoretical framework to account for the architecture of actin networks from the dynamics of their assembly. We study the simplest situation of actin structures assembling *de novo* from a fixed number of initially short filaments, mirroring existing *in vitro* experiments[4,5,7], as well as,

for example, cytoskeletal reassembly in a newly formed bleb[13], and actin recovery following drug treatment[14]. We consider a system of polymerizing and diffusing filaments that tend to bundle irreversibly when they come into contact (Fig. 1a), as observed experimentally[15]. Bundling can however be sterically blocked by the presence of other filaments (Fig. 1b) that becomes increasingly likely as the filaments elongate. At early times, the filaments are very short and diffusion is fast. Bundling thus proceeds unimpeded by steric constraints, and the number of bundles in the solution decreases over time as thicker bundles are formed through the merging of thinner ones. As a result, blocking becomes less likely and further bundling events are facilitated in a positive feedback mechanism. As the filaments grow, diffusion slows down and bundles come into contact more rarely. As a consequence, blocking finally outpaces reaction and the system becomes kinetically arrested. This basic mechanism allows us to formulate simple, experimentally testable scaling predictions for the bundle size and concentration, including an abrupt change in system behaviour upon kinetic trapping. We numerically validate these predictions in simulations of rod-like bundles over five orders of magnitude in concentration and four orders of magnitude in filament growth velocity, a much broader range than is accessible to existing detailed simulations[16]. We thus develop a robust, easily extendable framework to describe the nonequilibrium physics of cytoskeletal network assembly.

## Results

**A model for kinetic arrest based on filament entanglement**. We model actin bundles as impenetrable, infinitely thin, rigid[17] rods in a homogeneous solution of crosslinkers. These rods grow at a constant velocity $v$, and their diffusion coefficient is given by $D \sim k_BT/\eta L$ in the Rouse approximation[18], where $L = vt$ is the length of a filament and $\eta$ is the viscosity of the surrounding solution. When two rods come within a distance $b$ of the order of the size of a crosslinker, they react with a rate $k$ to merge into a rod-like bundle as in Fig. 1a. Note that the chemical rate constant $k$ is associated with the crosslinker binding rate and not the filament merging time. Indeed, the latter is much shorter than the typical filament reaction time, as further detailed in the discussion. Although nonmerged bundles may be connected by a few crosslinkers, such connections are short-lived ($\sim 1$ s for $\alpha$-actinin[12]) and we neglect them over the timescale of minutes involved in network formation. As a result, kinetic trapping in our model arises from steric entanglement between densely packed rods. This scenario is consistent with experimental evidence that entanglement can induce kinetic trapping in actin networks even in the absence of crosslinkers[19].

**Filament interactions involve several dynamical regimes**. We first develop a mean-field approach considering a homogeneous solution of isotropically oriented rods of concentration $c$. This concentration accounts for bundles of any thickness, including single filaments, which we see as 'one-filament bundles', and evolves according to

$$\frac{dc}{dt} = -r(c,L)c, \qquad (1)$$

where $L = vt$ and $r(c, L)$ is the rate with which one rod bundles with any other.

In dilute systems, where the average distance between two rods is much larger than their length ($cL^3 \ll 1$), $r(c, L)$ is effectively due to a two-body interaction. We denote it by $r^{(2)}$ and estimate it separately in the case of reaction-limited and diffusion-limited systems. In a reaction-limited system, two rods within interaction range $b$ bundle at a rate $k$. As the probability for a rod to be

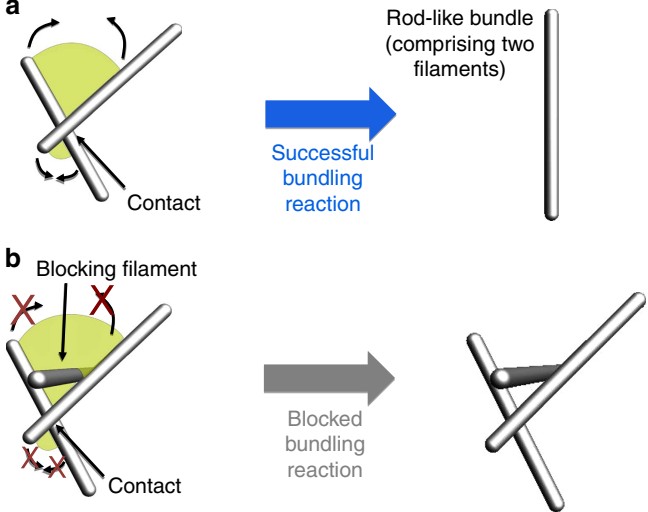

**Figure 1 | Basic mechanisms of filament network assembly.** (**a**) When two actin filaments come into contact, they attempt a bundling reaction (thin arrows). If there are no filaments in the surroundings, the attempt results in a single bundle. (**b**) If other filaments are found on the path of the bundling reaction (green surface), bundling is blocked because of steric interaction (red crosses).

within interaction range of another is $\sim cbL^2$, the total two-body rate of bundling is:

$$r_{\text{react}}^{(2)} = A_r kcbL^2, \qquad (2)$$

where $A_r$ is a dimensionless prefactor of order one. In the diffusion-limited case, the orientation of the rods can rotationally diffuse over the whole sphere in a much shorter time than it takes them to come into contact with one another. The rate with which one rod encounters any other through diffusion is thus given by $r_{\text{diff}}^{(2)} \sim cDL$ (ref. 20). Therefore:

$$r_{\text{diff}}^{(2)} = A_d c \frac{k_B T}{\eta}, \qquad (3)$$

where $A_d$ is another dimensionless prefactor. As evidenced by the different $L$-dependences of $r^{(2)}$ in equations (2 and 3), while the rods grow, diffusion slows down relative to reaction and the dynamics transition from reaction limited to diffusion limited. This happens at a critical length $L \sim L_c = \sqrt{k_B T / \eta kb}$.

In concentrated systems ($cL^3 \gtrsim 1$) bundling is affected by the presence of surrounding rods, yielding a rate:

$$r(c, L) = r^{(2)}(c, L)\left[1 - P_b\left(cL^3\right)\right], \qquad (4)$$

where the rate $r^{(2)}$ at which bundling is attempted is given by equations (2 and 3) and $P_b$ is the probability for the attempt to be *blocked* as in Fig. 1b. To determine the blocking probability, we note that $1 - P_b = (1 - p_b)^{N-2}$, where $p_b$ is the probability for an individual, randomly placed rod to block the attempt, and $N$ is the total number of rods in the system. To estimate $p_b$, we consider two rods with tangent vectors $\hat{n}$ and $\hat{n}'$ coming into contact at their midpoints. Denoting with $V$ the volume of the system, with $\hat{n}''$ the orientation of the third, potentially blocking rod and letting $\alpha(\hat{n}, \hat{n}')L^2/2$ be the area of the bundling path pictured in Fig. 1b, the probability that the third rod intersects this path reads $p_b = \int d^2\hat{n}'' \alpha(\hat{n}, \hat{n}')L^3|(\hat{n} \times \hat{n}') \cdot \hat{n}''|/2V$. In the thermodynamic limit ($N, V \to \infty$) this yields $1 - P_b(cL^3) = \int_0^{\pi/2} d\theta \sin\theta \exp(-\pi cL^3 \theta)$, which we plot in Fig. 2b. The blocking probability $P_b$ becomes large at large concentration $c$, accounting for the experimental observation that although

bundling speeds up with increasing $c$ at low $c$ (because of binary collisions), the opposite trend is observed at higher concentrations (when three or more body blocking becomes predominant)[4].

**Mean-field dynamical scenarios and final morphologies**. We now use our model to predict the final structure of a system of filaments. As shown in Fig. 2a, four different scenarios can develop, depending on the initial bundle concentration $c_0$, on the reaction rate $k$ and on the polymerization velocity $v$. As $L(t=0) = 0$, the system always starts off in the $cL^3 \ll 1$ reaction-limited regime, implying, through equations (1 and 2):

$$c(t) = \frac{c_0}{1 + (t/\tau_r)^3}, \qquad (5)$$

with $\tau_r \sim 1/(c_0 kbv^2)^{1/3}$. This solution predicts a crossover from $c = c_0$ for $t \ll \tau_r$ to $c \propto t^{-3}$ for $t \gg \tau_r$. Scenario (1) (Fig. 2a, topmost line) applies to slow-reacting filaments ($kb < v$), for which blocking happens before this first transition. The concentration $c$ thus never departs from its initial value $c_0$: a homogeneous network of single filaments of concentration $c_0$ is formed.

For fast-reacting filaments ($kb > v$) blocking takes over at a time larger than $\tau_r$. Bundles thus form, and three possible scenarios can develop depending on $c_0$. Scenario (2) (Fig. 2a, second line from the top) describes cases where substantial bundling takes place before the system transitions from reaction limited to diffusion limited, that is, $\tau_r < \tau_c = L_c/v$, or equivalently $c_0 > c_c = \frac{v}{kb}\left(\frac{\eta kb}{k_B T}\right)^{3/2}$. Equation (5) is thus valid for $t < \tau_c$, and the $c \propto t^{-3}$ decay is valid for $\tau_r < t < \tau_c$. As a result $cL^3$ remains constant while the rods grow, thus staving off blocking as long as $t < \tau_c$. At $\tau_c$, the system becomes diffusion limited, and equations (1 and 3) imply that the concentration decays as $c \sim \eta/k_B Tt$ as long as $cL^3 \ll 1$. Blocking then induces kinetic arrest for $cL^3 \sim 1$, implying $L \sim L_b = \sqrt{k_B T/\eta v}$, or equivalently $t \sim \tau_b = \sqrt{k_B T/\eta v^3}$, yielding a final concentration $c_f \sim c_b = \eta/k_B T\tau_b = (\eta v/k_B T)^{3/2}$ independent of the initial concentration $c_0$. Scenario (3)

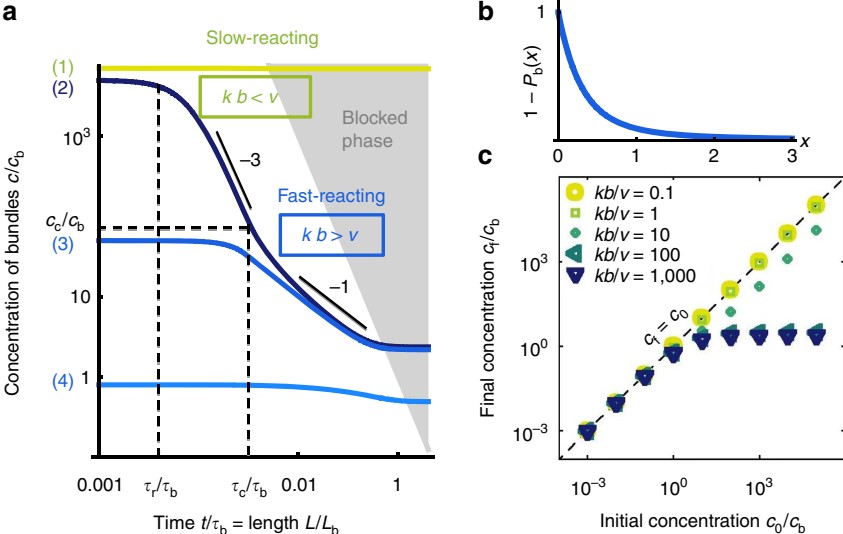

**Figure 2 | Dynamical evolution of our mean-field model.** (**a**) Numerical solutions of equations (1–4) for the four different scenarios (1) through (4) described in the main text. Here the system transitions from reaction limited to diffusion limited through $r^{(2)} = r_{\text{react}}^{(2)}$ for $L < L_c$ and $r^{(2)} = r_{\text{diff}}^{(2)}$ for $L \geq L_c$. An alternative, smoother interpolation $1/r^{(2)} = 1/r_{\text{react}}^{(2)} + 1/r_{\text{diff}}^{(2)}$ yields similar curves. Here $kb/v = 0.1$ for slow-reacting filaments (light grey (green) curve), and $kb/v = 1,000$ for fast-reacting filaments (dark grey (blue) curves). The grey region materializes the condition $cL^3 > 1$, where blocking is observed. (**b**) Probability $1 - P_b(cL^3)$ that an attempted bundling event does not get blocked. (**c**) Final bundle concentration $c_f$ as a function of $c_0$. The dashed line materializes the final concentration of a homogeneous network of single filaments ($c_f = c_0$).

(Fig. 2a, third line from the top) is relevant for $c_b < c_0 < c_c$. In this regime $\tau_c < \tau_r$ and the bundle concentration thus transitions from its initial plateau directly to the $c \propto t^{-1}$ diffusive regime. As in the previous scenario, blocking occurs at $t = \tau_b$, resulting in a final concentration of the order of $c_b$. Finally, scenario (4) (Fig. 2a, bottommost line) is relevant for $c_0 < c_b$. In that case, by the time the filaments get into contact through diffusion at $t \sim \eta/k_B T c_0$ the system is already concentrated, and most bundling attempts are therefore blocked, yielding $c_f \sim c_0$.

Our results regarding the final morphology of the networks are summarized in Fig. 2c: in low-concentration ($c_0 < c_b$) and/or slow-reacting ($kb \lesssim v$) systems, no bundling takes place. The final state is a homogeneous network of single filaments of concentration $c_f \sim c_0$. In contrast, in fast-reacting, high-concentration systems ($c_0 > c_b$ and $kb > v$), the system evolves to a network of bundles with concentration $c_f \sim c_b$ independent of $c_0$ and a characteristic number of filaments per bundle equal to $c_0/c_f \sim c_0/c_b$. For $c_0 \geq c_b$, going from a slow- to a fast-reacting system produces an abrupt shift from a homogeneous network of filaments to a network of bundles with a concentration lower by several orders of magnitude.

**Brownian dynamics simulations.** To assess the validity of our homogeneous, isotropic, mean-field dynamical scenarios, we conduct numerical simulations of our model. We simulate a solution of initially very short, randomly oriented, impenetrable rods and implement their growth as well as their standard Brownian dynamics with diffusion coefficients $D_{||} = k_B T/\eta L$, $D_\perp = D_{||}/2$ for their longitudinal and transverse translation, respectively, and $D_r = 6D_{||}/L^2$ for their rotation[18]. For each time step, the algorithm assesses the probability for each rod to react with its closest neighbour (the 'target' rod) that is assumed to be fixed (the fixed rod is itself moved in a separate step). To estimate this probability, we write the Fokker–Planck equation describing the stochastic dynamics of the distance $d$ between the two rods in the limit where $d \ll L$ (cases that violate this condition are benign in practice, as they yield a negligible reaction probability anyway):

$$\partial_t P(d,t) = D_\perp \partial_d^2 P(d,t) - 2kb\delta(x)P(d,t). \quad (6)$$

Here $P(d,t)$ is the probability distribution of $d \in [0, +\infty)$, and the right-hand side includes a sink term representing reactions between the two rods in the limit of a very short interaction range $b$. Equation (6) assumes the convention $\int_0^\infty \delta(x)dx = 1/2$. This yields a probability of reaction between the two rods initially separated by $d_0$ over one time step $dt$ of the simulation:

$$P_{attempt} = \mathrm{erfc}\left(\frac{d_0}{2\sqrt{D_\perp dt}}\right) - e^{\frac{kb(d_0 + kbdt)}{D_\perp}}\mathrm{erfc}\left(\frac{d_0 + 2kbdt}{2\sqrt{D_\perp dt}}\right). \quad (7)$$

Note that equation (7) is and must be fully valid even in cases

where $d_0$ is of the order of the typical diffusion and reaction length scales $\sqrt{D_\perp dt}$ and $kbdt$. The algorithm implements a bundling attempt with a probability $P_{attempt}$.

In the case where bundling is indeed attempted, the algorithm determines if any blocking rod is present in the bundling path (Fig. 1). If there is one, bundling is aborted and the attempting filament is moved in close proximity to the closest blocking rod. If bundling is successful, the attempting rod is deleted, representing its merging with the target rod. In the case where bundling is not attempted, diffusion proceeds as in normal Brownian dynamics, although with a reflecting boundary between rods to ensure their impenetrability[21]. Note that a single blocking rod can never derail bundling if it is not itself entangled with the rest of the network. Indeed, if the blocking rod is free to move and bundle with the attempting rod, they will do so within a few diffusion steps and the bundling of the two first rods will then be allowed to proceed. Thus, the transition to kinetic arrest is a true many-body effect in our simulations.

Snapshots of the resulting dynamics are shown in Fig. 3 at times corresponding to the four successive regimes of scenario (2). The evolution of the concentration in our simulations confirms the four predicted scenarios for the evolution of the rod concentration (Fig. 4a). Consistent with Fig. 2c, they also show that the final morphologies are either homogeneous networks of single filaments or strongly bundled phases, with an abrupt transition from one to the other (Fig. 4b) as a high-concentration system goes from slow reacting to fast reacting. The final structures do not display significant overall orientational order (Fig. 4c), consistent with both our model and *in vitro* observations[4,5,7].

The good agreement between our theory and simulations comes with one quantitative difference. Because of the more complex geometry of the simulations, kinetic arrest there sets in for a relatively large value of $cL^3$ ($\simeq 10$), allowing more time for bundling and thus driving the final concentration down. As seen in Fig. 4a, this delayed blocking reveals an additional dynamical regime with a slope steeper than $-1$. In this regime, the rod crosses over to a faster-than-diffusive exploration of space thanks to its ballistic polymerization, leading to a speed-up of bundling before blocking (see Methods and Fig. 6). Note however that this regime can never fully develop if blocking is present, and thus that the scaling scenarios described above remain valid in our simulations, although with modified prefactors.

## Discussion

The cytoskeleton of living cells is fundamentally out of equilibrium, and is constantly shaped by two major active processes: the operation of embedded molecular motors, and the constant self-assembly of its components. Although the statistical

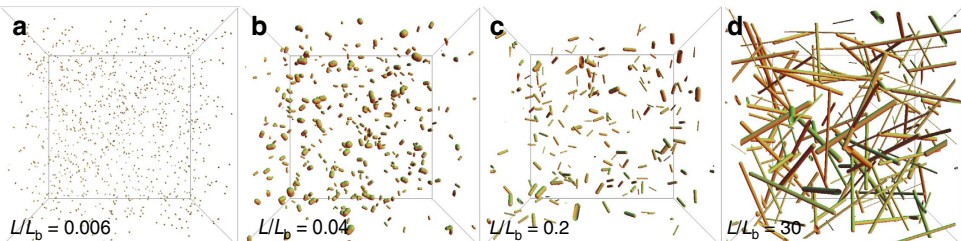

**Figure 3 | Snapshots of our Brownian dynamics simulations.** The state of the system in each of the four different regimes identified in our mathematical model are illustrated: (**a**) initial plateau $c \propto t^0$, (**b**) reaction-limited regime $c \propto t^{-3}$, (**c**) diffusion-limited regime $c \propto t^{-1}$ and (**d**) blocked regime $c \propto t^0$. To facilitate visualization despite order of magnitude changes in filament concentrations, each of the four panels is taken from a different simulation with a different box size, ensuring that between 100 and 1,000 rods are visible in each picture. The corresponding parameters and absolute concentrations can be read off from the white-filled symbols in Fig. 4a. In this figure the cross-sectional area of each rod is proportional to the number of filaments within the bundle.

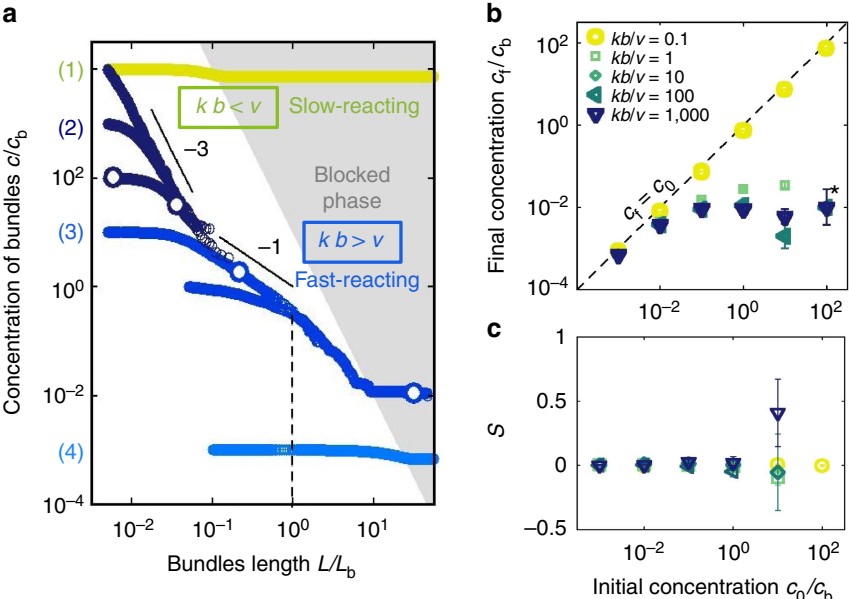

**Figure 4 | Dynamical evolution of our Brownian dynamics simulations.** (**a**) Concentration of solutions of $N = 10^3$ rods as a function of their length for $kb/v = 0.1$ (light grey (green)) and $kb/v = 1,000$ (dark (blue) symbols). The initial concentration ($c_0$) and filament length ($L_0$) for each simulation can be read on the graph, and the white-filled symbols indicate the data points used for the snapshots of Fig. 3. The grey region materializes the condition $cL^3 > 10$, where blocking is observed. (**b**) Final bundle concentration as a function of $c_0$ for $N = 10^4$ rods of initial length $L_0/L_b = 0.1$. Error bars give an estimate of the relative uncertainty on the number of remaining filaments at the end of the simulation $\left[ \delta \ln(c_f) = \delta N_{final}/N_{final} = 1/\sqrt{N_{final}} \right]$. For the most highly concentrated, fast-reacting conditions investigated ($c_0/c_b = 10^2$ and $kb/v \geq 1$, marked with an asterisk), bundling is so strong that all rods in our simulations collapse into one, terminating the dynamics for reasons independent of blocking. (**c**) Scalar nematic order parameter $S = \langle 3((\hat{n}^{(\alpha)} \cdot \hat{v})^2 - 1) \rangle$ for the data of (**b**) following blocking. In the definition of $S$ the index $\alpha$ refers to the $\alpha$-th rod, and $\hat{v}$ is the eigenvector corresponding to the largest eigenvalue of the tensorial order parameter $Q_{ij} = 3/2 \sum_{\alpha,\beta} (\hat{n}_i^{(\alpha)} \hat{n}_j^{(\beta)} - \delta_{ij}/3)$ (ref. 32). Error bars show the s.e.m. associated with the determination of the average involved in the computation of $S$, again indicating the error associated with small filament numbers. The nematic order parameter is not computed for final states where only one rod is present (namely, for the data points with $c_0/c_b = 10^2$, $kb/v \geq 1$).

mechanics of the former is the subject of a substantial experimental and theoretical literature[22], our understanding of the collective dynamics induced by the latter is very limited. Inspired by recent experiments, we introduce a versatile theoretical framework to investigate this problem, based on rate equations supplemented with a mean-field, entanglement-induced kinetic trapping term. Brownian dynamics simulations validate our theoretical assumptions, and show that our results are robust to changes in the detailed interactions between bundles.

We analyse our model in a simple situation consistent with existing *in vitro* experiments[4,5,7]. Although quantitative comparisons are impeded by technical challenges in resolving single filaments and thin bundles in these specific studies, our main qualitative predictions are all paralleled by the data. Bundle densities thus vary over orders of magnitude upon changes in the initial filament concentration $c_0$, and the timescale required for their formation decreases sharply upon an increase of $c_0$ (ref. 4). This is reminiscent of our predicted transition from the slowly relaxing ($c \propto t^{-1}$ at early times) scenario (3) at low $c_0$ to the faster ($c \propto t^{-3}$) scenario (2) at larger $c_0$. An increasing crosslinker concentration (analogous to an increase in $k$ in the model) further induces a sharp transition from a homogeneous (scenario (1)) to a bundled network (scenario (2) or (3)). An additional 10-fold increase in crosslinker concentration however hardly modifies the mesh size of the network, strongly reminiscent of our abrupt transition from a slow-reacting to a fast-reacting system of fixed concentration $c_b$ for $c_0 > c_b$ (ref. 4). Our model also predicts that an increased reaction rate is equivalent to a decreased polymerization velocity through the dimensionless parameter $kb/v$. Consistent with this, in ref. 5 an increase in $v$

through the use of the formin mDia1 causes the final bundle concentration to rapidly increase, then plateau out. More quantitatively, refs 4,5 use crosslinker α-actinin at concentrations of the order of $c_\alpha \approx 2 \, \mu M$. Given the α-actinin–actin binding rate $k_{on} = 5 \, \mu M^{-1} s^{-1}$ (ref. 23), we estimate that two actin filaments within an interaction distance $b \approx 30$ nm (the size of an α-actinin molecule) bind with a rate $k = k_{on} c_\alpha = 10 \, s^{-1}$. For $v = 10^{-2} \, \mu m \, s^{-1}$, this yields $kb/v \approx 30$ for the typical initial actin filament concentration $c_0 \approx 0.1 \, \mu M$. This is consistent with the formation of bundles observed under the aforementioned experimental conditions, and suggests that those *in vitro* assays can indeed transition between scenarios (1), (2) and (3) as their parameters are varied. We moreover predict $\tau_c \approx 370 \, s$ and $\tau_b \approx 2,000 \, s$, comparable to the observed gelation time $t \approx 600 \, s$.

These quantitative estimates further allow a discussion of the domain of validity of our model's main assumptions. We first discuss our approximation that the merging between two bundles is instantaneous. In general, the time required to merge two filaments is the sum of the time for the two filaments to find each other and form their first crosslink, plus a time $\tau_m$ required to complete their merging. Direct measurements[15] indicate that the latter timescale is of the order of a few hundred milliseconds at most. (This number is measured in the presence of large beads that slow down the merging dynamics because of hydrodynamic friction; the merging timescale $\tau_m$ is probably significantly smaller in the situation considered here, where such beads are not present.) This timescale is much shorter than the typical evolution timescales $\tau_c$ and $\tau_b$ evaluated above. More quantitatively, we estimate in Methods that the delay $\tau_m$ to merging will have negligible effects on the final concen-

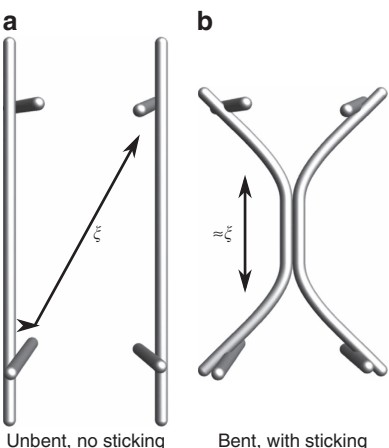

**Figure 5 | Assessment of the bundles' propensity to bend.** We consider two vertical bundles each blocked by the rest the network, with typical mesh size $\xi$ (**a**). Thermal fluctuations or internal network stresses may cause the two filaments to deform and come into contact (**b**). Denoting by $\epsilon$ the binding free energy associated with the binding of a crosslinker and by $\ell$ the typical distance between crosslinks, we assess whether such configurations are energetically favourable by comparing the energy bonus $\approx \xi\epsilon/\ell$ due to crosslinker binding to the energy penalty $\approx k_B T L_p/\xi$ because of filament bending, where $L_p$ is the persistence length of the bundle. The latter exceeds the former for mesh sizes smaller than $\sqrt{k_B T L_p \ell/\epsilon}$, ($\simeq 3\,\mu m$ with the parameters of the main text). Our treatment of bundles as rigid is thus justified in these networks.

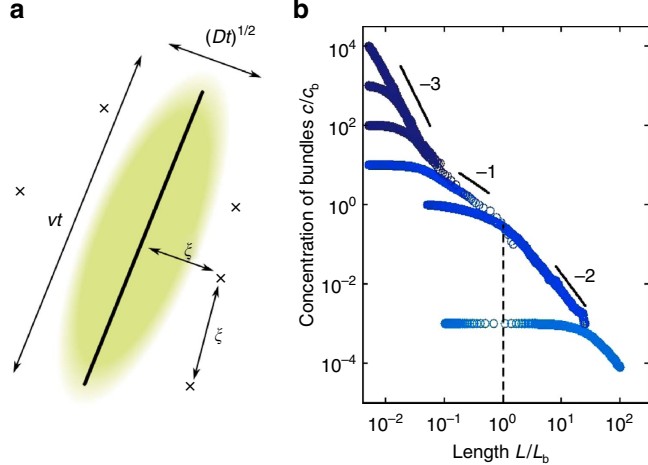

**Figure 6 | Characterization of the $L > L_b$ scaling regime in a system without blocking.** (**a**) Schematic of the situation considered for our scaling argument, showing the rod of interest (solid line), the area explored by it in a time $t$ (coloured region) and the other rods intersecting the plane of the figure (crosses). (**b**) Evolution of the concentration in a Brownian dynamics simulation identical to that of Fig. 4a with blocking turned off. The predicted $-2$ slope regime is clearly visible for $L > L_b$. The absence of blocking induces fast, unhindered decay of the filament concentrations towards zero, attesting to the importance of blocking for the stabilization of the cytoskeletal morphologies discussed above.

tration provided that $\tau_m \ll c_0^{-1/3}\nu^{-1} \simeq 25\,s$, confirming the merging time can indeed be neglected.

Our model also neglects actin bending and crosslinking, treating bundles as rigid rods throughout their dynamics. To assess the domain of validity of this approximation[24], we compare the energetic incentive for two bundles entangled with the rest of the network to merge over a fraction of their length and compare it with the bending cost of doing so (Fig. 5). Guided by the detailed simulations of ref. 25, we consider bundles of $N \simeq 10$ filaments with persistence length $L_p \approx N^2 \ell_p$, where $\ell_p \simeq 10\,\mu m$ is the actin persistence length, and assume that merging the two bundles brings a free energy bonus $\epsilon \simeq 4k_B T$ per crosslinker with a typical spacing between crosslinks of $\ell \simeq b \simeq 30\,nm$. As shown in Fig. 5, these parameters imply that filament bending will become prevalent for network mesh sizes of the order of $\sqrt{k_B T L_p \ell/\epsilon} \simeq 3\,\mu m$. This estimate is in line with the final morphologies observed in ref. 25, where bundles are bent on length scales of the order of one to two µm, comparable to the network mesh size. This estimate suggests that networks with smaller mesh sizes, which include most cellular structures as well as the structure-defining early stages of our simulations, will undergo only moderate bundle bending, compatible with our approach. In contrast, networks with larger mesh sizes, including those studied in some *in vitro* assays, will undergo significant bending and deformation, and may therefore not be well described by our model. These deformations could facilitate the collapse of entangled structures towards a more energetically favourable (that is, more crosslinked) state. This could compromise the mechanical integrity of these networks and account for the formation of inhomogeneous structures with large gaps as observed in refs 26,27.

The good agreement between our predictions and the experiments of refs 4,5 suggests that topological entanglement between filaments could be the major driver of kinetic arrest in cytoskeletal systems. Depending on the system considered,

its action in regulating the thickness of actin bundles could be complemented by other mechanisms. For instance, the build-up of elastic strains[28] has been proposed to regulate the width of bundles crosslinked by the very short crosslinker fascin[29], although the importance of this mechanism is less clear in the α-actinin bundles used in refs 4,5. Other effects ignored here, for example, transient sticking between unbundled filaments or the effective increase in length incurred by a bundle upon coalescence with another, may thus not be essential to gain a first understanding of the resulting network structures. Such effects could however easily be included in our framework if warranted by more precise experimental comparisons, as will the physiologically important effects of spontaneous filament nucleation or the coexistence of several crosslinker types with different bundling behaviours[30]. Detailed simulations will also be useful in assessing the influence of the addition of these and other experimentally relevant features to our model. Although our current solution-like model does not explicitly describe the network's mechanical properties, it does predict its typical mesh size and bundle thickness, whose relationship to the network's mechanical response has been the subject of substantial modelling efforts[31]. Finally, further experimental and theoretical work is needed to elucidate the network structure in the biologically relevant presence of depolymerization/severing that could give rise to fundamentally nonequilibrium steady states.

Overall, our study provides a first theoretical account of the nonequilibrium mechanisms responsible for the actin structures observed *in vivo* and *in vitro*. It further illustrates that these dynamical processes can lead to sharp transitions between dramatically different network structures, hinting that cells need only harness relatively modest changes in their internal composition to generate the large variety of morphologies that characterize the cytoskeleton.

## Methods

**Speed-up of bundling before blocking.** Here we rationalize the speed-up of bundling observed in our Brownian dynamics simulations just before the system

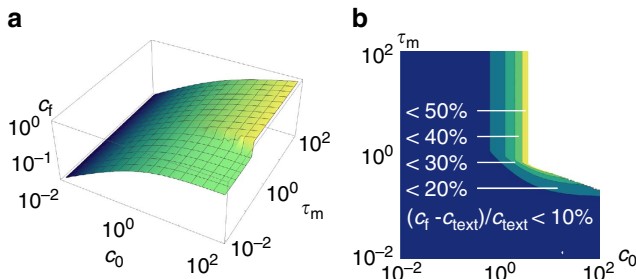

**a**

$c_f$

**b**

$\tau_m$

< 50%
< 40%
< 30%
< 20%
$(c_f - c_{text})/c_{text} < 10\%$

$c_0$

**Figure 7 | Quantitative effect of delayed bundling on $c_f$. (a)** Final concentrations as given by equation (13) in absolute value. (**b**) Corresponding deviations relative to the $\tau_m = 0$ prediction (denoted as $c_{text}$). Such violations are substantial only in the regime where $\tilde{c}_0 > 1$ and $\tilde{\tau}_m < \tilde{c}_0^{-1/3}$.

becomes blocked in Fig. 4a. This speed-up is a signature of the system crossing over to a new scaling regime for $r^{(2)}$ as $L$ becomes larger than $L_b$, that is, as the longitudinal growth of the rod becomes faster than its longitudinal diffusion. In practice, this regime has little incidence on our model as bundling becomes hindered by blocking precisely at $L = L_b$ (in our mean-field discussion) or shortly thereafter (in our simulations). Here we present a scaling argument showing that $c \propto t^{-2}$ in this regime, and display numerical evidence to that effect.

Let us consider a rod of interest lying in the plane of Fig. 6a. As the rod diffuses and grows, it encounters other rods that intersect the plane of the figure, and attempts to bundle with them. In an homogeneous, isotropic solution, the typical distance between two rods is $\xi \sim (cL)^{-1/2}$ (ref. 31). The typical number of other rods encountered by the rod of interest after a time $t$ is $n(t) \sim A(t)/\xi^2$, with $A(t)$ the typical area of the plane of the figure visited by the rod of interest within a time $t$. The width of this area is of the order of $\sqrt{Dt}$, with $D \sim k_B T/(\eta L)$ the typical diffusion coefficient of the rod (representing a combination of transverse and rotational diffusion). Here we consider a diffusing and growing rod with length $L \gg L_b$, implying that the rate of longitudinal diffusion of the rod (inducing a displacement $\sim \sqrt{Dt}$) is negligible in front of its growth rate (inducing a displacement $\sim vt$). As a result the length of the area grows as $vt$ and $A(t) \sim \sqrt{Dt} \times vt$. Combining these expressions and using $L = vt$ we find that $n(t) \sim (k_B T/\eta)^{1/2} v^{3/2} ct^2$.

The typical reaction rate $r^{(2)}$ is the rate at which the rod of interest encounters other rods, yielding $r^{(2)} \sim \mathrm{d}n/\mathrm{d}t \sim (k_B T/\eta)^{1/2} v^{3/2} ct$. Applying equation (1) in the absence of blocking we thus find

$$\frac{\mathrm{d}c}{\mathrm{d}t} = -r^{(2)} c \sim -\left(\frac{k_B T}{\eta}\right)^{1/2} v^{3/2} c^2 t, \qquad (8)$$

which yields in the long-time asymptotic limit

$$c \sim \frac{\eta^{1/2}}{(k_B T)^{1/2} v^{3/2} t^2}, \qquad (9)$$

This $c \propto t^{-2}$ scaling is indeed observed in our simulations in the absence of blocking, as shown in Fig. 6b.

**Incidence of delayed bundling on the final concentration.** Here we assess the effect of a finite bundle merging time $\tau_m$ on the final rod concentration. We place ourselves in the diffusion-limited regime that, as described in Results, is the important one when considering the transition to blocking. The rate of bundling attempts is thus

$$r_{\mathrm{diff}}^{(2)} = -\frac{k_B T}{\eta} c. \qquad (10)$$

To account for the additional hindrance to bundling, we assume that two rods that come into contact at a time $t$ only complete their bundling at time $t + \tau_m$. During this time interval, the two unbundled rods are linked together and thus diffuse as a single object, implying that they count as a single rod in estimating the concentration that enters the attempt rate of equation (10). However, as they are not yet bundled they retain their full blocking power towards other bundling events until $t + \tau_m$. Denoting by $c(t)$ the number of independently diffusing objects at time $t$, the full mean-field bundling rate thus reads:

$$r(t) = -\frac{k_B T}{\eta} c(t)\{1 - P_b[c(t - \tau_m)L^3]\}, \qquad (11)$$

as the concentration of rods relevant for blocking at time $t$ is identical to the concentration of free rods at time $t - \tau_m$. Finally, here we assume that the bundling dynamics becomes abruptly blocked as $cL^3$ exceeds one (note that this approximation preserves all the scaling results derived in the Results section).

Rescaling time by $\tau_b = \sqrt{k_B T/\eta v^3}$ and concentration by $c_b = (\eta v/k_B T)^{3/2}$, the full dimensionless concentration equation becomes

$$\frac{\mathrm{d}\tilde{c}}{\mathrm{d}\tilde{t}} = -\tilde{c}^2\{1 - H[\tilde{c}(\tilde{t} - \tilde{\tau}_m)\tilde{t}^3]\}, \qquad (12)$$

where $\tilde{c} = \tilde{c}_0$ for $\tilde{t} \leq 0$ and $H$ is the Heaviside step function. The final solution of this equation displays two regimes, depending on whether kinetic arrest takes over before or after the first bundling event is completed:

$$\tilde{c}_f = \begin{cases} \tilde{c}_0/\left(1 + \tilde{c}_0^{2/3}\right) & \text{if } \tilde{c}_0\tilde{\tau}_m^3 > 1 \\ \left(\frac{\sqrt{81x^2 - 12} + 9x}{18}\right)^{1/3} + \left(\frac{2/3}{\sqrt{81x^2 - 12} + 9x}\right)^{1/3} & \text{if } \tilde{c}_0\tilde{\tau}_m^3 \leq 1 \end{cases}, \qquad (13)$$

with $x = 1/\tilde{c}_0 - \tilde{\tau}_m$.

These final concentrations are plotted in Fig. 7, along with their relative deviation from the result at $\tau_m = 0$. In practice, our results are insensitive to the value of $\tau_m$ as long as $\tilde{\tau}_m < \tilde{c}_0^{-1/3} \Leftrightarrow \tau_m < c_0^{-1/3} v^{-1}$, as discussed in the Results section.

**Data availability.** The computer code used for this study as well as the data generated and analysed are available from the corresponding author on request.

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

## Acknowledgements

We thank Emmanuel Trizac for stimulating discussions. This work was supported by grants from Université Paris-Sud's *Attractivité* and CNRS' *PEPS-PTI* programs, Marie Curie Integration Grant PCIG12-GA-2012-334053, 'Investissements d'Avenir' LabEx PALM (ANR-10-LABX-0039-PALM), ANR Grant ANR-15-CE13-0004-03 and ERC Starting Grant 677532. The group of M.L. belongs to the CNRS consortium CellTiss.

## Author contributions

M.L. designed the research. G.F., N.L. and M.L. carried out the analytical calculations. G.F. carried out the numerical simulations. G.F. and M.L. wrote the paper.

## Additional information

**Competing financial interests:** The authors declare no competing financial interests.

**Publisher's note**: 

