## [Peer Review File · Nature Communications]

Reviewers' comments:

Reviewer #1 (Remarks to the Author):

The authors theoretically and computationally investigate the question of bundle formation in solutions of stiff biopolymers such as actin and keratin. They propose an interesting kinetic mechanism to explain the different observed bundle sizes in different experimental conditions. However, the model seems to rely on a non trivial assumption that - in my view - has not been adequately proven, and the sensitivity of the model predictions to this assumption has not been investigated (see below).

My main concern is that the model seem to rely on the assumption that once two filaments form one cross-link, they proceed to form a fully connected bundle, and the main kinetic impediment for large bundle formation are other intervening filaments that prevent the "zipping" of two contacting filaments into a bundle. This assumption is far from obvious, and may be true - but might not be. It could be probed in direction simulations, but the simulations in the paper also seem to rely on the same assumption, instead of testing its validity.

Therefore, at this point, I cannot recommend the paper for publication in Nature Communications.

Reviewer #2 (Remarks to the Author):

The manuscript by Foffano et al. presents a theory for the formation of a non-equilibrium state of polymerizing filaments in the presence of linker bundling proteins. In the final state, filament bundling has stopped because steric hindering by a third filament prevents two filaments in contact to bundle, a form of constrained equilibrium. The theory is supported by a numerical simulation.

The model is clean, the calculations are clear and simple, and the comparison with the simulation is direct. I do have three comments concerning the assumptions on which the model is based.

1) The rate k with which two filaments rotate to form a bundle after initial contact is assumed to be independent of filament length L . If this rate is limited by viscous drag of the rotating filaments then k should decrease with L as $1/L$. This would alter the L dependence in the reaction-limited regime. k also will depend on the thickness of the bundle.

2) My next comment concerns the competition between the steric blocked state, as proposed in this manuscript, and the equilibrium state, which is a single fat bundle. Suppose two filaments are blocked, at a given point in time, by a third filament along the lines proposed by the paper. A kind of Brownian ratchet could gradually work the blocking filament out from in between the two filaments: if a thermal fluctuation would cause the blocking filament to move in a way that the subtended angle between the other two filaments could become smaller then the resulting gain in adhesion energy between the filaments would act like a rectifier and prevent the reverse fluctuation. By not allowing this by fiat, the authors do not provide us with a sense when to expect the system to achieve a thermal equilibrium state and when not. Note that the simulation also does not seem to allow for the ratchet mechanism.

3) The model focuses on the free energy gain of bundling multiple filaments into a thicker bundle but explicitly does not allow linkers to connect individual filaments into a network, or a network of bundles. The reviewer is co-author on a brownian dynamics simulation of the bundling of semi-flexible filaments of fixed length in the presence of reversible linkers:

Rheology of Semiflexible Bundle Networks with Transient Linkers

Kei W. Müller, Robijn F. Bruinsma, Oliver Lieleg, Andreas R. Bausch, Wolfgang A. Wall, and Alex J. Levine

Phys. Rev. Lett. 112, 238102, 2016

The system ended up in a non-equilibrium network of bundles composed of linked filaments held together by other linker molecules at intersections between the bundles. Other simulation studies report similar states (e.g., PHYSICAL REVIEW E 89, 062602 (2014)). Experimental studies of actin in the presence of linker molecules also report linked networks of bundles. (O. Pelletier, E. Pokidysheva, L. S. Hirst, N. Bouxsein, Y. Li, and C. R. Safinya
Phys. Rev. Lett. 91, 148102)

This type of nonequilibrium state, with a frozen network structure, differs from the more entropic state proposed by the authors in the model, where there are constant structural fluctuations that are prevented from forming a bundle by steric hindering.

I am not demanding that the authors start from scratch with a more complex model. What I do want is a clear physical discussion by the authors in the concluding section discussing competition between their final state with both the equilibrium fat bundle state and the cross-linked network state. Under what conditions do they expect their non-equilibrium state to dominate. Finally, the authors may want to comment on the fact that there is evidence (Claesens et al. PNAS 2008 vol. 105 no. 26 8819–8822) that actin bundles may have a finite, quasi-equilibrium thickness because of chiral frustration, which has been the subject of a number of theory papers by Grason et al.

Minor comments: Figure 1 top right should show two filaments. I am irritated by the use of jargon borrowed from control theory ("feed forward") in a physics paper but that is not important.

REVIEWERS' COMMENTS:

Reviewer #1 (Remarks to the Author):

I have no further objections.

Reviewer #2 (Remarks to the Author):

I am satisfied by the response of the authors and recommend publication.

The dynamics of filament assembly define cytoskeletal network morphology

Reply to the reviewers

Foffano *et al.*

October 11, 2016

We thank both reviewers for their careful reviews and positive comments concerning the interest of our proposed mechanism and the cleanliness of our treatment, respectively. As noted by them, our manuscript proposes a simple mechanism for a very complex biological process. To highlight this mechanism and draw its essential predictions, we employ an undeniably simplified model. A careful discussions of the validity of our approximations is thus important to our point.

Consistent with this, both reviewers have very legitimately called for this discussion to be expanded. We have done so following the second reviewer's advice to not "start from scratch", but to provide a "clear physical discussion [...] in the concluding section". This discussion also goes to address the first reviewer's main concern on the validity of our assumptions. Our text modifications are highlighted in blue in the revised text and reproduced below.

We feel that the reviewers' comments helped to significantly strengthen our discussion, and hope that they will deem our revised manuscript suitable for publication.

R1 First Reviewer

The authors theoretically and computationally investigate the question of bundle formation in solutions of stiff biopolymers such as actin and keratin. They propose an interesting kinetic mechanism to explain the different observed bundle sizes in different experimental conditions. However, the model seems to rely on a non trivial assumption that - in my view - has not been adequately proven, and the sensitivity of the model predictions to this assumption has not been investigated (see below).

My main concern is that the model seem to rely on the assumption that once two filaments form one cross-link, they proceed to form a fully connected bundle, and the main kinetic impediment for large bundle formation are other intervening filaments that prevent the "zipping" of two contacting filaments into a bundle. This assumption is far from obvious, and may be true - but might not be. It could be probed in direction simulations, but the simulations in the paper also seem to rely on the same assumption, instead of testing its validity.

Therefore, at this point, I cannot recommend the paper for publication in Nature Communications.

We thank the reviewer for her or his comment, and for acknowledging that our proposed mechanism is plausible, but not obvious; this is in our opinion the optimal place to be doing theory. She or he implies that our main hypothesis should be “proven” prior to our study. We are pessimistic regarding the potential of detailed, “direct” numerical simulations in reliably providing such a proof, as such studies invariably involve the introduction of many parameters and (often implicit) assumptions whose influence on the final result tends to be difficult to assess. In particular detailed simulations are typically very bad at dealing with widely separated time scales; such time scales are indeed present in the systems we study, ranging from zipping and reaction times of the order of a few dozen milliseconds to a blocking time scale $\tau_b \approx 2000$ s (see estimates in the text). For these reasons we would rather seek such a proof from experiments, and the point of our work is precisely to formulate experimentally testable predictions that will be used to this effect. That said we agree that detailed simulations can provide important insights into the influence of features not explicitly included in our model, as we now explicitly acknowledge in the revised text:

“Detailed simulations will also be useful in assessing the influence of the addition of these and other experimentally relevant features to our model.” (page 7)

While detailed simulations are well beyond the scope of our current study, we do acknowledge that we can and should do a better job at presenting the existing experimental evidence supporting our main hypothesis. As discussed in our reply to reviewer 2, we also extended our discussion to connect with existing detailed simulations in one fruitful case (in page 6; see below). Here we discuss each of the three parts of the reviewer’s restatement of our assumptions:

1. Two contacting filaments tend to spontaneously bundle (“zip”).
2. Intervening filaments impede bundling.
3. This impediment is more important than other effects in preventing zipping.

Here we discuss the existing evidence for each point. We also discuss the sensitivity of our model to them as requested by the reviewer; this point is the subject of a new section of our Supporting Information.

1. *Two contacting filaments tend to spontaneously bundle (“zip”).* Filament zipping is directly observed experimentally. Streichfuss *et al.* (2011) thus monitor the zipping of two filaments held by optical tweezers in the presence of an inter-filament interaction on the order of $\approx 5k_B T/100$ nm. This interaction is smaller than or comparable with the one associated with α -actinin crosslinking, ($\approx 10k_B T$ per crosslink with more than one crosslink per 100 nm), suggesting a similar or faster zipping dynamics in the presence of crosslinks. In this experimental study, this dynamics is moreover slowed down by hydrodynamic friction as the filament ends are bound to micron-sized beads. Despite this, zipping still occurs within a few hundred milliseconds. The situation considered in our paper does not involve any such bead, implying that zipping should be much easier and quicker.

Sensitivity: Our results are obviously sensitive to the presence of zipping. This is indeed implicitly tested in our study, as forbidding zipping is equivalent to setting $k = 0$ in our description. The case of such a very small k is addressed by the two top curves of Figs. 2 and 3, and leads to a homogeneous network of single filaments.

This experimental basis for our assumption is now mentioned in the text:

“We consider a system of polymerizing and diffusing filaments that tend to bundle irreversibly when they come into contact [Fig. 1(A)], as observed experimentally [Streichfuss *et al.* (2011)]” (page 1)

2. *Intervening filaments impede bundling.* Indirect experimental evidence of this effect is provided, *e.g.*, in Fig. 2f of Falzone *et al.* (2012). There the typical rate of filament bundling is shown to depend non-monotonically on the initial filament concentration. This is consistent with our model where two-body interactions (binary collisions) promote bundling, accounting for an increase of the bundling rate with increasing concentration c at small c , while three- or more-body interactions hinder it (through entanglements), accounting for the decrease of the bundling rate at high c .

Sensitivity: Fig. S1 of our study shows that this assumption is essential to our results; if three-body interactions are removed the system always finds its fully bundled ground state.

This point is now discussed our manuscript through the following additions:

“The blocking probability P_b becomes large at large concentration c , accounting for the experimental observation that while bundling speeds up with increasing c at low c (due to binary collisions), the opposite trend is observed at higher concentrations (when three- or more-body blocking becomes predominant) [Falzone *et al.* (2012)].” (page 2)

“The absence of blocking induces fast, unhindered decay of the filament concentrations towards zero, attesting to the importance of blocking for the stabilization of the cytoskeletal morphologies discussed in the main text.” (Supporting information)

3. *This impediment is more important than other effects in preventing zipping.* This is in our opinion the most interesting and potentially controversial part of our proposal, whose validity will *in fine* be assessed by putting our predictions to the test of experiments. Assessing its plausibility requires establishing a list of other potential kinetic impediments to large bundle formation following a first contact between bundles, and the associated time scales. While drawing an exhaustive list is impossible, obvious other kinetic impediments include the detachment of one or both of the coalescing filaments from other bundles to which they might be crosslinked but not bundled ($\tau \simeq 1 \text{ s}^{-1}$ for the crosslinker α -actinin), and the zipping time (faster than 100 ms according to Streichfuss *et al.* (2011) and the above discussion).

Sensitivity: As discussed in the revised main text the delays to bundling discussed here are much shorter than both the crossover time τ_c and the blocking time τ_b , suggesting that they are indeed negligible. Since this important point is indeed deserving of a sensitivity analysis as suggested by the reviewer, we now examine the question in more detail in a new section of our Supporting Information (not reproduced here). We again conclude that these other impediments to bundling are negligible, as discussed in the main text:

“[...] We moreover predict $\tau_c \approx 370$ s and $\tau_b \approx 2000$ s, comparable to the observed gelation time $t \approx 600$ s.

These quantitative estimates further allow a discussion of the domain of validity of our model’s main assumptions. We first discuss our approximation that the merging between two bundles is instantaneous. In general, the time required to merge two filaments is the sum of the time for the two filaments to find each other and form their first crosslink, plus a time τ_m required to complete their merging. Direct measurements [Streichfuss *et al.* (2011)] indicate that the latter time scale is of the order of a few hundred milliseconds at most^a. This time scale is much shorter than the typical evolution time scales τ_c and τ_b evaluated above. More quantitatively, we estimate in the Supporting Information that the delay τ_m to merging will have negligible effects on the final concentration provided that $\tau_m \ll c_0^{-1/3} v^{-1} \simeq 25$ s, confirming the merging time can indeed be neglected.” (page 6)

^aThis number is measured in the presence of large beads which slow down the merging dynamics due to hydrodynamic friction; the merging time scale τ_m is probably significantly smaller in the situation considered here, where such beads are not present.

Overall we believe that while they do not provide a definite proof, existing experimental evidence and simple estimates lend sufficient support to our basic hypothesis to justify our exploration of its implications and potential experimental signatures. Our results will in turn guide further *in vitro* and *in silico* experiments in the search for the mechanisms of actin network self-assembly.

R2 Second Reviewer

The manuscript by Foffano et al. presents a theory for the formation of a non-equilibrium state of polymerizing filaments in the presence of linker bundling proteins. In the final state, filament bundling has stopped because steric hindering by a third filament prevents two filaments in contact to bundle, a form of constrained equilibrium. The theory is supported by a numerical simulation.

The model is clean, the calculations are clear and simple, and the comparison with the simulation is direct. I do have three comments concerning the assumptions on which the model is based.

1) The rate k with which two filaments rotate to form a bundle after initial contact is assumed to be independent of filament length L . If this rate is limited by viscous drag of the rotating filaments then k should decrease with L as $1/L$. This would alter the L dependence in the reaction-limited regime. k also will depend on the thickness of the bundle.

The short answer to the reviewer’s question is that the rate k is not limited by viscous drag. More specifically, the reviewer’s question is based on an interpretation of k as the inverse of the merging time τ_m (identical to the “zipping time” discussed above), *i.e.*, the inverse of the time required for bundling once a crosslink has formed between two bundles. In fact this time scale is typically negligible, and k^{-1} is the time for the first crosslink to bind two nearby bundles, which has no reason to depend on L . We have made this point clearer in the text:

“The chemical rate constant k is associated with the crosslinker binding rate and not the filament merging time. Indeed, the latter is much shorter than the typical filament reaction time, as further detailed in the discussion.” (page 2)

Our rationale for neglecting the merging time is the subject of the new section of the Supplementary Information, and is summarized in the third item of our response to reviewer 1.

2) My next comment concerns the competition between the steric blocked state, as proposed in this manuscript, and the equilibrium state, which is a single fat bundle. Suppose two filaments are blocked, at a given point in time, by a third filament along the lines proposed by the paper. A kind of Brownian ratchet could gradually work the blocking filament out from in between the two filaments: if a thermal fluctuation would cause the blocking filament to move in a way that the subtended angle between the other two filaments could become smaller then the resulting gain in adhesion energy between the filaments would act like a rectifier and prevent the reverse fluctuation. By not allowing this by fiat, the authors do not provide us with a sense when to expect the system to achieve a thermal equilibrium state and when not. Note that the simulation also does not seem to allow for the ratchet mechanism.

The reviewer is correct in pointing out that under low density conditions a single filament trapped between two zipping filaments could be pushed around, thus allowing zipping to proceed. At high density however, this third filament will be entangled with and immobilized by additional surrounding filaments, preventing this process. Therefore one lonely entanglement can never stop bundling; only sufficient entanglement of the blocking filaments with its neighbors, and of these neighbors with their own neighbors *etc.* yields network-wide kinetic arrest. In other words kinetic arrest is a collective, not a local phenomenon. Elucidating these collective aspects is important to understand when the system will achieve a thermal equilibrium state.

While the specific mechanism discussed by the reviewer is not explicitly taken into account in our simulations, another closely related one is. In the spirit of the reviewer’s comment, this effect prevents artificial situations where a single blocking filament derails a bundling event altogether. To understand this mechanism, consider a situation where two filaments (denoted by 1 and 2 in the following) attempt bundling, but are blocked by filament 3. Following the failed bundling attempt, our simulation moves filament 2 very close to filament 3, guaranteeing that 2 and 3 will collide and attempt bundling within a few diffusion steps. If filament 3 is indeed unhindered by further filaments in the network, this attempt will result in the coalescence of 2 and 3, removing the blocking and allowing the bundling with filament 1 to proceed. If the coalescence between 2 and 3 is prevented by filament 4, filament 3 will similarly be moved close to 4, *etc.*. Therefore the simulation does tell us whether an initial blocking event will be eventually resolved into a relaxation to equilibrium and considers the whole cascade of possible interactions with blocking filaments in the spirit of the reviewer’s comment. This is a crucial test of our analytical approach, where many-body positional correlations are neglected.

We added a few sentences to the text to clarify this situation:

“Note that a single blocking rod can never derail bundling if it is not itself entangled with the rest of the network. Indeed, if the blocking rod is free to move and bundle with the attempting rod, they will do so within a few diffusion steps and the bundling of the two first rods will then be allowed to proceed. Thus the transition to kinetic arrest is a true many-body effect in our simulations.” (page 4)

3) The model focuses on the free energy gain of bundling multiple filaments into a thicker bundle but explicitly does not allow linkers to connect individual filaments into a network, or

a network of bundles. The reviewer is co-author on a brownian dynamics simulation of the bundling of semi-flexible filaments of fixed length in the presence of reversible linkers:

Rheology of Semiflexible Bundle Networks with Transient Linkers

Kei W. Müller, Robijn F. Bruinsma, Oliver Lieleg, Andreas R. Bausch, Wolfgang A. Wall, and Alex J. Levine

Phys. Rev. Lett. 112, 238102, 2016

The system ended up in a non-equilibrium network of bundles composed of linked filaments held together by other linker molecules at intersections between the bundles. Other simulation studies report similar states (e.g., PHYSICAL REVIEW E 89, 062602 (2014)). Experimental studies of actin in the presence of linker molecules also report linked networks of bundles. (O. Pelletier, E. Pokidysheva, L. S. Hirst, N. Boussein, Y. Li, and C. R. Safinya Phys. Rev. Lett. 91, 148102)

This type of nonequilibrium state, with a frozen network structure, differs from the more entropic state proposed by the authors in the model, where there are constant structural fluctuations that are prevented from forming a bundle by steric hindering.

I am not demanding that the authors start from scratch with a more complex model. What I do want is a clear physical discussion by the authors in the concluding section discussing competition between their final state with both the equilibrium fat bundle state and the cross-linked network state. Under what conditions do they expect their non-equilibrium state to dominate.

We agree with the reviewer that some crosslinking will take place between bundles in the network's final state. If the bundles are very stiff, only a few crosslinks will bind and the resulting bond will be very transient (of the order of the α -actinin bond lifetime ≈ 1 s), implying a similar behavior as in our model. On the other hand, if the bundles are sufficiently flexible, the crosslinks will deform them and zip them together, similar to the morphologies observed in the reviewer's work [Müller *et al.* (2014)]. The simulations of Pandolfi *et al.* (2014) (*i.e.*, the reviewer's second reference) appear to display a transition of this type, and we added a reference to their work in the revised manuscript. We also added a forgotten reference to Pelletier *et al.* (2003) (the third reference), which pioneered the type of experiments discussed in our manuscript.

We also added a paragraph to the main text estimating which regime is expected depending on the physical parameters of the network. Our criterion is consistent with the observation of significant bending and crosslinking between bundles in Müller *et al.* (2014), but predicts that our model is applicable to the physically distinct early-time dynamics with short bundles and small mesh sizes that is most crucial to the network self-organization described in our manuscript:

Figure R1: Assessment of the bundles' propensity to bend. We consider two vertical bundles each blocked by the rest of the network, with typical mesh size ξ (left). Thermal fluctuations or internal network stresses may cause the two filaments to deform and come into contact (right). Denoting by ϵ the binding free energy associated with the binding of a crosslinker and by ℓ the typical distance between crosslinks, we assess whether such configurations are energetically favorable by comparing the energy bonus $\approx \xi\epsilon/\ell$ due to crosslinker binding to the energy penalty $\approx k_B T L_p/\xi$ due to filament bending, where L_p is the persistence length of the bundle. The latter exceeds the former for mesh sizes smaller than $\sqrt{k_B T L_p \ell/\epsilon}$, ($\simeq 3 \mu\text{m}$ with the parameters of the main text). Our treatment of bundles as rigid is thus justified in these networks.

“Our model also neglects actin bending and crosslinking, treating bundles as rigid rods throughout their dynamics. To assess the domain of validity of this approximation [Pandolfi *et al.* (2014)], we compare the energetic incentive for two bundles entangled with the rest of the network to merge over a fraction of their length and compare it to the bending cost of doing so (Fig. R1). Guided by the detailed simulations of Ref. [Müller *et al.* (2014)], we consider bundles of $N \simeq 10$ filaments with persistence length $L_p \approx N^2 \ell_p$, where $\ell_p \simeq 10 \mu\text{m}$ is the actin persistence length, and assume that merging the two bundles brings a free energy bonus $\epsilon \simeq 4k_B T$ per crosslinker with a typical spacing between crosslinks of $\ell \simeq b \simeq 30 \text{ nm}$. As shown in Fig. R1, these parameters imply that filament bending will become prevalent for network mesh sizes of the order of $\sqrt{k_B T L_p \ell/\epsilon} \simeq 3 \mu\text{m}$. This estimate is in line with the final morphologies observed in Ref. [Müller *et al.* (2014)], where bundles are bent on length scales of the order of one to two micrometers, comparable to the network mesh size. This estimate suggests that networks with smaller mesh sizes, which include most cellular structures as well as the structure-defining early stages of our simulations, will undergo only moderate bundle bending, compatible with our approach. By contrast, networks with larger mesh sizes, including those studied in some *in vitro* assays, will undergo significant bending and deformation, and may therefore not be well described by our model. These deformations could facilitate the collapse of entangled structures towards a more energetically favorable (*i.e.*, more crosslinked) state. This could compromise the mechanical integrity of these networks and account for the formation of inhomogeneous structures with large gaps as observed in Refs. [Lieleg *et al.* (2009); Nguyen & Hirst (2011)].” (page 6)

Finally, the authors may want to comment on the fact that there is evidence (Claesens et al. PNAS 2008 vol. 105 no. 26 8819-8822) that actin bundles may have a finite, quasi-equilibrium

thickness because of chiral frustration, which has been the subject of a number of theory papers by Grason et al.

We thank the reviewer for pointing out this additional mechanism of bundle size control, which we now also discuss:

“Depending on the system considered, its action in regulating the thickness of actin bundles could be complemented by other mechanisms. For instance, the build-up of elastic strains [Grason (2015)] has been proposed to regulate the width of bundles crosslinked by the very short crosslink fascin [Claessens *et al.* (2008)], although the importance of this mechanism is less clear in the α -actinin bundles used in Refs. [Falzone *et al.* (2012, 2013)].” (page 6)

Minor comments: Figure 1 top right should show two filaments.

We agree that this figure represents a bundle made of two filaments. However our initial intention in drawing a single cylinder was to make clear that bundles containing two (or more) filaments are treated as a single object in our model. We attempted to prevent the potential for confusion pointed out by the reviewer while keeping this point clear by changing the figure text from “rod-like bundle” to “rod-like bundle (comprising two filaments)”.

I am irritated by the use of jargon borrowed from control theory (“feed forward”) in a physics paper but that is not important.

We strive to avoid jargon and use language suited to a physics paper. We replaced “feed forward” with “positive feedback” in the hope that this will address the reviewer’s concern while still conveying the idea. We of course welcome any alternative suggestion.

References

- Claessens, M M A E, Semmrich, C, Ramos, L, & Bausch, A R. 2008. Helical twist controls the thickness of F-actin bundles. *Proc Natl Acad Sci U S A*, **105**(26), 8819–22.
- Falzone, Tobias T, Lenz, Martin, Kovar, David R, & Gardel, Margaret L. 2012. Assembly Kinetics Determine the Architecture of α -actinin Crosslinked F-actin Networks. *Nat. Commun.*, **3**(May), 861.
- Falzone, Tobias T., Oakes, Patrick W., Sees, Jennifer A., Kovar, David R., & Gardel, Margaret L. 2013. Actin Assembly Factors Regulate the Gelation Kinetics and Architecture of F-actin Networks. *Biophys. J.*, **104**(8), 1709–1719.
- Grason, Gregory M. 2015. Colloquium: Geometry and optimal packing of twisted columns and filaments. *Rev. Mod. Phys.*, **87**(2), 401.
- Lieleg, Oliver, Schmoller, Kurt M, Cyron, Christian J, Luan, Yuxia, Wall, Wolfgang A, & Bausch, Andreas R. 2009. Structural polymorphism in heterogeneous cytoskeletal networks. *Soft Matter*, **5**(9), 1796–1803.

- Müller, Kei W., Bruinsma, Robijn F., Lieleg, Oliver, Bausch, Andreas R., Wall, Wolfgang A., & Levine, Alex J. 2014. Rheology of Semiflexible Bundle Networks with Transient Linkers. *Phys. Rev. Lett.*, **112**(23), 238102.
- Nguyen, Lam T, & Hirst, Linda S. 2011. Polymorphism of highly cross-linked F-actin networks: Probing multiple length scales. *Phys. Rev. E*, **83**(3), 031910.
- Pandolfi, Ronald J, Edwards, Lauren, Johnston, David, Becich, Peter, & Hirst, Linda S. 2014. Designing highly tunable semiflexible filament networks. *Phys Rev E Stat Nonlin Soft Matter Phys*, **89**(6), 062602.
- Pelletier, O., Pokidysheva, E., Hirst, L. S., Bouxsein, N., Li, Y., & Safinya, C. R. 2003. Structure of Actin Cross-Linked with Alpha-Actinin: A Network of Bundles. *Phys. Rev. Lett.*, **91**(14), 148102.
- Streichfuss, Martin, Erbs, Friedrich, Uhrig, Kai, Kurre, Rainer, Clemen, Anabel E.-M., Böhm, Christian H. J., Haraszti, Tamás, & Spatz, Joachim P. 2011. Measuring Forces between Two Single Actin Filaments during Bundle Formation. *Nano Lett.*, **11**(9), 3676–3680.